# OpenReview forum: "Learning to Watermark LLM-generated Text via Reinforcement Learning"
_ICLR.cc/2025/Conference — Submitted to ICLR 2025_

### Official Review · Reviewer_CSmp · 2024-10-29

**Soundness:** 2
**Presentation:** 3
**Contribution:** 2
**Rating:** 5
**Confidence:** 4

**Summary:**

This paper proposes an LLM watermarking method that injects the watermark at the LLM fine-tuning stage, by training the LLM jointly with another detector language model via RL methods, such that the LLM's output can be reliably detected by the detector model.

**Strengths:**

- The paper tackles an important problem of watermarking open-source LLM models to detect their output.

- The paper is relatively well-written and clear.

**Weaknesses:**

- The success of the proposed method may be highly dependent on the training dataset and process (including several hyperparameters), which brings to question whether it can be generalized well in practice (please see questions below).

- The paper should also be more thorough in its discussion of related works, benchmarks and claimed contributions. For example, as it is a training-based watermarking approach, the paper should at least include a discussion on other related fine-tuning watermarking methods (e.g. backdoor watermarking, like [1]).

  [1]  Li, Shen, et al. "Double-I Watermark: Protecting Model Copyright for LLM Fine-tuning."

- Also, as it is comparing with training-free watermarking methods, it should also discuss and include training-free watermarking methods that can be applied on open-source models, such as [2], especially since the paper explicitly highlights open-sourcing watermarked models as a problem of existing works.

   [2]  Lau et al. "Waterfall, Framework for Robust and Scalable Text Watermarking and Provenance for LLMs"


-  The proposed method does not seem to provide the significant advantages in runtime overhead as claimed in Sec 5.5. For example, compared to the KGW method for the Llama2 model, the advantage in generation time is small (e.g. <0.01s or <2%) while disadvantage in detection time is relatively much larger (~0.02s or ~64%). This is on top of the very significant training overheads incurred by the method, which may not be 1-time given the requirements for additional adversarial training for better results, as described in various parts of the paper.

- The claim that verification runtime is less important than generation runtime should also be more calibrated, as in practice fast verification runtime may be more important because in some settings active screening needs to be done to check for watermarks which will involve checking through large corpuses of text.

- While the authors claim that compared to the RLHF pipeline, the only additional cost is "training an extra reward model", this may not be the case -- the more complex objective could impact the training dynamics and convergence of the model, and this potential impact should be discussed.


- The sequential-code watermarking approach (Sec. 4.4) involving exact match (line 245) has the significant weakness of being very brittle to perturbations -- any small changes to the text may cause a non-match of the sequence and cause detectability to drop to 0. The comparison with other statistical-based model-centric watermarking method is not appropriate as those cases do not require an exact code sequence match, unlike the proposed method.

- Given the claim from the paper that the method can be used to watermark open-source models and detect their output, the watermarking detection performance for OOD tasks would be important. I suggest that the authors shift up the OOD section in the appendix, and provide further details ad elaboration on this aspect.

**Questions:**

- Please provided additional explanations on any convergence analysis of the proposed iterative training algorithm.

- Please elaborate on the differences between the proposed approach and other finetuning-based watermarking approaches, such as backdoor watermarking [1].

- For the main results, as the proposed method involves finetuning with the training data, a fair comparison would involve the various methods applied to the fine-tuned model on the same training data. Otherwise, it would be hard to interpret any evaluation made on the utility or performance of the watermarked model. Please provide some empirical results on this.

- Please elaborate on specifically what training data was used and what test dataset was used to evaluate the model performance, especially since the proposed method will have access to the training data while the baselines do not. Would the prompts used during testing need to be similar to the training dataset? The appendix on OOD tasks seem to imply that.

- The proposed method seems to face significant challenges with substitution attack for the Llama2 setting on the C4 dataset. Could the authors please elaborate on reasons behind this, and whether it is potentially indicative of the sensitivity of the method's success to the hyperparameters used for training?

- Please provide some sensitivity results, if available, on how the empirical results may change depending on the various key hyperparameters that need to be set for the proposed method

- Given that training is required for the proposed method, the utility of the model may be of concern. The evaluations done are on tasks where there is high degree of tolerance for error (e.g. perplexity scores). Table 10 in the appendix seems to indicate that the response 'without watermark' and 'with watermark' feature relatively different semantic meaning. Have the authors done any experiments where standardized benchmarks tests are done on the watermarked LLMs, with results compared to the LLMs of the baseline methods?

---

> ### Author Response · Authors · 2024-11-20
> **Response to Reviewer CSmp [1/2]**
>
> We thank the reviewer for the valuable feedback. We provide our responses to the questions below.
>
> **Convergence analysis** (Please provided additional explanations on any convergence analysis of the proposed iterative training algorithm.)
>
> **Response**: We iteratively train the LLM and the detector to minimize the detection loss as shown in Equation 2. Note that both the LLM and the detector training goals are to decrease the detection loss. Since the two goals are aligned (rather than adverse), the process can be viewed as optimizing one goal over two parameter sets, and it is expected to converge to the minimal possible detection loss.
>
> **Related works** ([1] and [2] mentioned by the reviewer)
>
> **Response**: We thank the reviewer for pointing to these new categories of related works. We will clarify the difference here and also discuss them in the revision of the paper. First, our method is different with the backdoor watermarking methods ([1] mentioned by the reviewer) in that their methods focus on model watermarking, while our method focuses on output watermarking. The model watermarking methods will usually have a "trigger set", so that only prompts from this set will lead to different outputs that can be detected. On the other hand, our method requires that **all the output from the model will be detected by the detector**. Second, our method is different with the text-level watermarks ([2] mentioned by the reviewer) in that they focus on the watermarks injected into a piece of text, but not the LLM model. Although we can apply such methods to watermark LLM outputs in a post-hoc way, it requires extra generation overhead (in the case of [2], the time to run a 13B model for paraphrasing). We will include the discussion above in the revision of the paper.
>
> **Fair comparison** (For the main results, as the proposed method involves finetuning with the training data, a fair comparison would involve the various methods applied to the fine-tuned model on the same training data.)
>
> **Response**: We agree with the author that the finetuning on the PKU data could help with model performance on the safety score. We would like to argue that the finetuning on the C4 data is purely for watermark injection and does not have impact on model performance (i.e. perplexity). We will only regularize the watermarked model to have low KL divergence with the original model on the dataset, so the extra data will not improve the model performance.
>
> **Training setting** (Please elaborate on specifically what training data was used and what test dataset was used to evaluate the model performance, especially since the proposed method will have access to the training data while the baselines do not. Would the prompts used during testing need to be similar to the training dataset? The appendix on OOD tasks seem to imply that.)
>
> **Response**: In the main experiments, the training and testing datasets are from the same data distribution (C4 or PKU), but split in the standard way so that there are no identical data cases in training and testing sets. The experiments on OOD tasks evaluate the case where training and testing datasets are from different distributions.

---

> > ### Author Response · Authors · 2024-11-20
> > **Response to Reviewer CSmp [2/2]**
> >
> > **Robustness** (The proposed method seems to face significant challenges with substitution attack for the Llama2 setting on the C4 dataset. Could the authors please elaborate on reasons behind this, and whether it is potentially indicative of the sensitivity of the method's success to the hyperparameters used for training?)
> >
> > **Response**: We thank the reviewer for reminding us of the performance of the raw Llama model under substitution attacks. Instead of hyperparameter sensitivity, we owe it to the problem of adversarial robustness of data-driven models - a high-performance model on the original domain (benign C4 data) may have poor performance on a perturbed domain (C4 data with 20% randomly substituted words). On the other hand, we show the flexibility of our method that, when adversarial training is applied, our watermark can achieve high robustness and get close-to-perfect performance even when 50% words are randomly corrupted.
> >
> > **Hyper-parameter** (Please provide some sensitivity results, if available, on how the empirical results may change depending on the various key hyperparameters that need to be set for the proposed method)
> >
> > **Response**: The key hyperparameters of our method include the KL regularization coefficients $\beta, \gamma$ and reward tradeoff coefficients $\alpha$. We did not empirically observe a big impact of these factors on model performance within reasonable value ranges. We are conducting a full sensitivity analysis on these hyperparameters and will update the paper once ready.
> >
> > **Standardized benchmark** (Given that training is required for the proposed method, the utility of the model may be of concern. The evaluations done are on tasks where there is high degree of tolerance for error (e.g. perplexity scores). Table 10 in the appendix seems to indicate that the response 'without watermark' and 'with watermark' feature relatively different semantic meaning. Have the authors done any experiments where standardized benchmark tests are done on the watermarked LLMs, with results compared to the LLMs of the baseline methods?)
> >
> > **Response**: Currently, we evaluate the performance of LLM on the original task with perplexity (for C4) and safety score (for PKU). These are standard methods to evaluate the LLM performance after watermarking. We agree with the reviewer that the extra metrics are good ways to evaluate the effect of watermarks on original performance, and we plan to include them in future versions of the paper.

---

> > > ### Comment · Reviewer_CSmp · 2024-11-24
> > >
> > > Thank you for your responses. Some of the clarifications are useful, though several of my concerns are only partially addressed, e.g., issues such as the lack of some key empirical results and details/analysis, as the authors are aware. Hence, I will neither decrease nor increase my score, but maintain the current one.

---

### Official Review · Reviewer_CaER · 2024-11-01

**Soundness:** 2
**Presentation:** 3
**Contribution:** 2
**Rating:** 6
**Confidence:** 4

**Summary:**

The paper introduces a method for watermarking outputs from Large Language Models (LLMs) by embedding detectable signals into the model's weights rather than the text itself. This approach uses reinforcement learning to co-train the LLM and a paired detector, enhancing detection accuracy, robustness, and adaptability to new attacks. The method allows for open-sourcing watermarked models without releasing unwatermarked versions and incurs no generation overhead. The experiments demonstrate high detection rates and resilience to adversarial attacks while maintaining the model's utility.

**Strengths:**

- This paper introduces watermarking LLM by fine-tuning, which makes watermark detection easier and more robust to attacks such as paraphrasing.
- This manuscript is well-written and easy to follow.

**Weaknesses:**

- Fine-tuning the generative models and using an additional detector for watermark verification is not new, and related methods [1, 2] are supposed to be discussed in the related work section.
- I am concerned about the reliability of using a language model as the detector instead of statistical tests.
- It is unknown whether the extra fine-tuning process will introduce side effects or biases into LLMs, and there is no theoretical analysis of the changed parameters by additional fine-tuning.

Reference:

[1] Yu, Ning, et al. "Artificial fingerprinting for generative models: Rooting deepfake attribution in training data." Proceedings of the IEEE/CVF International conference on computer vision. 2021.

[2] Yu, Ning, et al. "Responsible disclosure of generative models using scalable fingerprinting." arXiv preprint arXiv:2012.08726 (2020).

**Questions:**

- The proposed method uses a learnable DNN as the detector, which can increase the watermark detection accuracy since both the LLM and the detector are optimized during watermarking. What if the adversary also provides a detector? That is to say, the adversary can keep the watermarked LLM fixed and optimize another LLM as his/her own detector. Note that the adversary does not need to modify the watermarked LLM, simply train another detector that can extract his/her watermark from the generated text, which would cast ambiguity over the verification process. How to deal with this situation?
- Will you release the source codes?

---

> ### Author Response · Authors · 2024-11-20
> **Response to Reviewer CaER**
>
> We thank the reviewer for the valuable feedback. We provide our responses below.
>
> **Adversarial attack** (The proposed method uses a learnable DNN as the detector, which can increase the watermark detection accuracy since both the LLM and the detector are optimized during watermarking. What if the adversary also provides a detector? That is to say, the adversary can keep the watermarked LLM fixed and optimize another LLM as his/her own detector. Note that the adversary does not need to modify the watermarked LLM, simply train another detector that can extract his/her watermark from the generated text, which would cast ambiguity over the verification process. How to deal with this situation?)
>
> **Response**: We thank the reviewer for pointing out a potential attack against our pipeline, where an adversary may train a surrogate watermark detector given the output of our watermarked LLM. We have two interpretations of the proposed attack based on the surrogate detector and will discuss them separately.
>
> In the first interpretation, the adversary may claim that the detector trained by him/her is the true detector and therefore casting ambiguity on which detector to trust during the verification. We think that this would not happen in common use cases. Usually, an LLM provider will inject the watermark and keep the detector as a secret. To detect the watermark, the LLM provider will apply the secret detector, so it is difficult for the adversary to add another detector in the pipeline.
>
> In the second interpretation, the adversary may launch an adversarial attack based on the surrogate model, so that a watermarked text can be repharsed to not be detected by the detector. We argue that this attack requires a costly adversarial attack on transformer-based language models, which is still ongoing research and we do not know, to the best of our knowledge, any widely recognized perfect attacks. Merely attacking the surrogate model is already difficult, as the AdvGLUE benchmark on language model robustness [a] shows that the average attack success rate on binary classification language models is only around 60%. The transferred attack success rate would be lower.
>
> [a] Wang, Boxin, et al. "Adversarial GLUE: A Multi-Task Benchmark for Robustness Evaluation of Language Models." Thirty-fifth Conference on Neural Information Processing Systems Datasets and Benchmarks Track (Round 2). 2021.
>
> **Code Opensourcing** (Will you release the source codes?)
>
> **Response**: Yes, we will open-source the code of our work when we publish our paper.

---

### Official Review · Reviewer_YCyk · 2024-11-02

**Soundness:** 3
**Presentation:** 3
**Contribution:** 2
**Rating:** 5
**Confidence:** 4

**Summary:**

This paper proposes a reinforcement learning-based watermarking method that simultaneously fine-tunes the model and trains a classifier to identify model-generated text. The method requires prompt-completion text pairs during detection and can be integrated with existing model alignment tasks. Experiments demonstrate good detectability and robustness against text modifications.

**Strengths:**

• Successfully proposes and implements a watermarking method using fine-tuning and reinforcement learning

• Conducts comprehensive experiments on watermark detectability and robustness

• Successfully integrates the proposed fine-tuning method into existing alignment workflows

**Weaknesses:**

• The method appears to require the prompt that generated the text being tested for watermarks. This prerequisite fundamentally differs from current inference-time watermarking methods. The authors don't explicitly discuss how this condition affects watermark embedding and detection

• The requirement of having the original prompt for detection significantly limits practical detection scenarios

• The detectability and robustness experiments don't explicitly discuss the impact of prompts. For example, it's unclear how changes to prompts might affect detectability

• The paper lacks details about baseline method parameters and settings, and these settings may not be comprehensive

**Questions:**

• Does the Detector D require prompt-completion text pairs as input? Does this mean the original prompt is needed during use? Doesn't this severely limit the watermark's practical applications?

• Why doesn't the no-fine-tuning method achieve the lowest perplexity? Intuitively, not fine-tuning the model should have minimal impact on generated text. Does this suggest that perplexity might not accurately reflect the watermark's impact on text quality?

• Since KGW changes its green list at each step, it naturally has poorer robustness. Would it be more fair to compare robustness with KGW family's unigram methods (“Provable robust watermarking for ai-generated text”)?

• What are the parameter settings for baseline methods in the experiments? For example, what are the size of green list and delta values in KGW?

• In the robustness experiments, did they only consider modifications to the generated text, or did they also examine how prompt modifications might affect watermark detection results?

**Details Of Ethics Concerns:**

Not applicable.

---

> ### Author Response · Authors · 2024-11-20
> **Response to Reviewer YCyk**
>
> We thank the reviewer for the valuable feedback. We provide our responses to the questions below.
>
> **Requirement of Original Prompt** (Does the Detector D require prompt-completion text pairs as input? Does this mean the original prompt is needed during use? Doesn't this severely limit the watermark's practical applications?)
>
> **Response**: We thank the reviewer for pointing out the requirement of original prompt in the paper. We would like to make two clarifications. First, we have the experiments of training and detecting the watermarks without the knowledge of original prompt in Appendix B.4 and Table 9. We show that without the knowledge of the prompt, our method can still achieve very high detection performance on C4 (AUC=0.9984) and PKU (AUC=0.9991). Second, we choose the setting of knowing the original prompt because it is reasonable in the prompt completion setting. Suppose the LLM is used to complete some prompt in the C4 news dataset, the resulting output to show to the public will more likely be the overall text (prompt + completion), rather than the completion-only text. We will make these two points clear in the main text of the paper.
>
> **Perplexity** (Why doesn't the no-fine-tuning method achieve the lowest perplexity?)
>
> **Response**: We thank the reviewer for referring to the perplexity numbers in the table. The perplexity metric is a commonly used metric in LLM watermark literatures (e.g. KGW / ITS papers) to evaluate the text quality of the watermark.  In the KGW paper, the PPL values have a wide range (3 to 13) and the text quality changes notably. In our case, perplexity is generally similar, and we view the small difference of perplexity as a result of randomness in text generation. The numbers show that our method will not affect text quality in general.
>
> **Baseline Implementations** (Would it be more fair to compare robustness with KGW family's unigram methods (“Provable robust watermarking for ai-generated text”); What are the parameter settings for baseline methods in the experiments? For example, what are the size of green list and delta values in KGW?)
>
> **Response**: We thank the reviewer for referring to the unigram methods. We viewed the provable robust watermark work as a variant of KGW, and therefore chose to discuss it in the related work but did not compare it in the experiments. According to Table 2 in their paper, the watermark still suffers from a notable performance drop under paraphrasing. For the baseline parameter setting, we use the default setting in their original implementation. Specifically, we use $\gamma=0.5$ and $\delta=2.0$ for KGW; we use $n=256$ for ITS and EXP. We will add the information in the revision of the paper.
>
> **Robustness evaluation** (In the robustness experiments, did they only consider modifications to the generated text, or did they also examine how prompt modifications might affect watermark detection results?)
>
> **Response**: We only made modifications to the generated texts but not the prompts. It is true that prompt modifications may also affect watermark detection. On the other hand, if we use the setting where the detector only sees the generated text (discussed in the text above, Appendix B.4 and Table 9), it will not affect the detection.

---

### Official Review · Reviewer_7Qwc · 2024-11-03

**Soundness:** 2
**Presentation:** 3
**Contribution:** 2
**Rating:** 3
**Confidence:** 3

**Summary:**

This paper explores embedding watermarks directly within large language models (LLMs) to detect and track misuse of generated content. Unlike prior token-level approaches, this research introduces a model-level watermark embedded into the model’s weights, detectable by a paired detector. The proposed method uses reinforcement learning to co-train the LLM and a detector, optimizing the LLM to produce watermarked text detectable with high accuracy while maintaining readability and model utility.

**Strengths:**

1. This paper proposes to fine-tune the LLMs to embed watermarks.
2. The proposed method is robust against different attacks.
3. The idea that combines the watermark embedding process with the alignment process is interesting.

**Weaknesses:**

1. The detection needs the original prompt, which is usually unavailable during the detection process.

2. This paper uses the D^{nw} (human-written prompt and answer) to fine-tune the LLM and detector. What I am worried about is that the detector learned the difference between human-written text and LLM-generated text instead of un-watermarked text (text generated by unwatermarked LLMs) and watermarked text. It would be good to present the results between the original LLM and the fine-tuned LLM, and try this watermarking method on some more powerful LLMs.

3. Could authors specify the model used to measure the PPL?

4. This watermarking method changed the parameter of the original LLM. I think it would be good to measure if this fine-tuning affects the performance of the original LLM using methods like FActScore, AlpacaFarm, etc.

**Questions:**

Please see above.

---

> ### Author Response · Authors · 2024-11-20
> **Response to Reviewer 7Qwc**
>
> We thank the reviewer for the valuable feedback. We provide our responses to the questions below.
>
> **Requirement of Original Prompt** (The detection needs the original prompt, which is usually unavailable during the detection process.)
>
> **Response**: We thank the reviewer for pointing out the requirement of original prompt in the paper. We would like to make two clarifications. First, we have the experiments of training and detecting the watermarks without the knowledge of original prompt in Appendix B.4 and Table 9. We show that without the knowledge of the prompt, our method can still achieve very high detection performance on C4 (AUC=0.9984) and PKU (AUC=0.9991). Second, we choose the setting of knowing the original prompt because it is reasonable in the prompt completion setting. Suppose the LLM is used to complete some prompt in the C4 news dataset, the resulting output to show to the public will more likely be the overall text (prompt + completion), rather than the completion-only text. We will make these two points clear in the main text of the paper.
>
> **Distinguishing between other LLMs** (This paper uses the D^{nw} (human-written prompt and answer) to fine-tune the LLM and detector. What I am worried about is that the detector learned the difference between human-written text and LLM-generated text instead of un-watermarked text (text generated by unwatermarked LLMs) and watermarked text.)
>
> **Response**: We appreciate the reviewer's concern on the possibility of detector's bias towards human vs. non-human texts. In Appendix A and Table 6, we evaluated the watermark detector on texts generated by other LLMs. We show that the detector can correctly classify the text by other LLMs as non-watermarked with high accuracy. In addition, we argue that this problem can be addressed with better training data selection. When we include the text generated by other LLMs in the training process, we can further improve the detector performance (denoted as "H+L" in Table 6).
>
> **PPL computation** (Could authors specify the model used to measure the PPL?)
>
> **Response**: We followed the KGW paper and used the OPT-2.7B model to measure the PPL.
>
> **Other model performance** (This watermarking method changed the parameters of the original LLM. I think it would be good to measure if this fine-tuning affects the performance of the original LLM using methods like FActScore, AlpacaFarm, etc.)
>
> **Response**: Currently, we evaluate the performance of LLM on the original task with perplexity (for C4) and safety score (for PKU). These are standard methods to evaluate the LLM performance after watermarking. We agree with the reviewer that the extra metrics are good ways to evaluate the effect of watermarks on original performance, and we plan to include them in future versions of the paper.

---

### Meta-Review · Area_Chair_FSFP · 2024-12-21

**Metareview:**

1x reject, 2x borderline reject, and 1x borderline accept. This paper proposes a reinforcement-learning-based approach to embed watermarks into the weights of large language models, accompanied by a detector that learns to identify watermarked outputs. The reviewers agree on the (1) idea of training both model and detector jointly, (2) extensive experiments on detection accuracy and robustness, (3) potential integration with existing alignment pipelines, and (4) clear and well-organized writing. However, they note (1) the requirement of having the original prompt for detection, (2) limited discussion on potential biases introduced by extra fine-tuning, (3) inadequate comparisons to other fine-tuning-based watermark approaches, and (4) concerns about whether the detector simply learns differences between human and LLM-generated text rather than watermarked vs. unwatermarked text. The authors did follow up with clarifications on detection without prompts and added references, yet the core concerns about biases, broader comparisons, and real-world practicality remain insufficiently addressed, so the AC leans to not accept this submission.

**Additional Comments On Reviewer Discussion:**

N/A

---

### Decision · Program_Chairs · 2025-01-22

Reject